META-RESEARCH

# Releasing a preprint is associated with more attention and citations for the peer-reviewed article

**Abstract** Preprints in biology are becoming more popular, but only a small fraction of the articles published in peer-reviewed journals have previously been released as preprints. To examine whether releasing a preprint on bioRxiv was associated with the attention and citations received by the corresponding peer-reviewed article, we assembled a dataset of 74,239 articles, 5,405 of which had a preprint, published in 39 journals. Using log-linear regression and random-effects meta-analysis, we found that articles with a preprint had, on average, a 49% higher Altmetric Attention Score and 36% more citations than articles without a preprint. These associations were independent of several other article- and author-level variables (such as scientific subfield and number of authors), and were unrelated to journal-level variables such as access model and Impact Factor. This observational study can help researchers and publishers make informed decisions about how to incorporate preprints into their work.

**DARWIN Y FU AND JACOB J HUGHEY\***

**\*For correspondence:**
jakejhughey@gmail.com

**Competing interests:** The authors declare that no competing interests exist.

## Introduction

Preprints offer a way to freely disseminate research findings while a manuscript undergoes peer review (*Berg et al., 2016*). Releasing a preprint is standard practice in several disciplines, such as physics and computer science (*Ginsparg, 2011*), and a number of organizations – including ASAPbio and bioRxiv.org (*Sever et al., 2019*) – are encouraging the adoption of preprints in biology and the life sciences. However, some researchers in these fields remain reluctant to release their work as preprints, partly for fear of being scooped as preprints are not universally considered a marker of priority (*Bourne et al., 2017*), and partly because some journals explicitly or implicitly refuse to accept manuscripts released as preprints (*Reichmann et al., 2019*). Whatever the reason, the number of preprints released each month in the life sciences is only a fraction of the number of peer-reviewed articles published (*Penfold and Polka, 2019*).

Although the advantages of preprints have been well articulated (*Bourne et al., 2017*; *Sarabipour et al., 2019*), quantitative evidence for these advantages remains relatively sparse. In particular, how does releasing a preprint relate to the outcomes – in so far as they can be measured – of the peer-reviewed article? Previous work found that papers posted on arXiv before acceptance at a computer science conference received more citations in the following year than papers posted after acceptance (*Feldman et al., 2018*). Another study found that articles with preprints on bioRxiv had higher Altmetric Attention Scores and more citations than those without, but the study was based on only 776 peer-reviewed articles with preprints (commensurate with the size of bioRxiv at the time) and did not examine differences between journals (*Serghiou and Ioannidis, 2018*). We sought to build on these efforts by leveraging the rapid growth of bioRxiv, which is now the

largest repository of biology preprints. Independently from our work, a comprehensive recent study has replicated and extended the findings of Serghiou and Ioannidis, although it did not quantify journal-specific effects or account for differences between scientific fields (*Fraser et al., 2019*).

## Results

We first assembled a dataset of peer-reviewed articles indexed in PubMed, including each article's Altmetric Attention Score and number of citations and whether it had a preprint on bioRxiv. (See Methods for full details. The code and data to reproduce this study are available on Figshare; see data availability statement below.) Because we sought to perform an analysis stratified by journal, we only included articles from journals that had published at least 50 articles with a preprint on bioRxiv. Overall, our dataset included 74,239 articles, 5,405 of which had a preprint, published in 39 journals between January 1, 2015 and December 31, 2018 (*Supplementary file 1*). Release of the preprint preceded publication of the peer-reviewed article by a median of 174 days (*Figure 1—figure supplement 2*).

Across journals and often within a journal, Attention Score and citations varied by orders of magnitude between articles (*Figure 1—figure supplements 3* and *4*). Older articles within a given journal tended to have more citations, whereas older and newer articles tended to have similar distributions of Attention Score. In addition, Attention Score and citations within a given journal were weakly correlated with each other (median Spearman correlation 0.18, *Figure 1—figure supplement 5*, and *Supplementary file 2*). These findings suggest that the two metrics capture different aspects of an article's impact.

We next used regression modeling to quantify the associations of an article's Attention Score and citations with whether the article had a preprint. To reduce the possibility of confounding (*Falagas et al., 2013*; *Fox et al., 2016*), each regression model included terms for an article's preprint status, publication date, number of authors, number of references, whether any author had an affiliation in the United States (by far the most common country of affiliation in our dataset, *Supplementary file 13*), whether any author had an affiliation at an institution in the 2019 Nature Index for Life

Sciences (a proxy for institutions that publish a large amount of high quality research), the last author publication age, and the article's approximate scientific subfield within the journal (*Supplementary file 4*). We inferred each last author's publication age (which is a proxy for the number of years the last author has been a principal investigator) using names and affiliations in PubMed (see Methods for details). We approximated scientific subfield as the top 15 principal components (PCs) of Medical Subject Heading (MeSH) term assignments calculated on a journal-wise basis (*Figure 1—figure supplements 6* and *7* and *Supplementary file 5*), analogously to how genome-wide association studies use PCs to adjust for population stratification (*Price et al., 2006*).

For each journal and each of the two metrics, we fit multiple regression models. For Attention Scores, which are real numbers, we fit log-linear and Gamma models. For citations, which are integers, we fit log-linear, Gamma, and negative binomial models. Log-linear regression consistently gave the lowest mean absolute error and mean absolute percentage error (*Figure 1—figure supplement 8* and *Supplementary file 6*), so we used only log-linear regression for all subsequent analyses (*Supplementary file 7*).

We used the regression fits to calculate predicted Attention Scores and citations for hypothetical articles with and without a preprint in each journal, holding all other variables fixed (*Figure 1*). We also examined the exponentiated model coefficients for having a preprint (equivalent to fold-changes), which allowed comparison of relative effect sizes between journals (*Figure 2*). Both approaches indicated higher Attention Scores and more citations for articles with preprints, although as expected Attention Score and citations showed large article-to-article variation (*Figure 1—figure supplement 9*). Similar to Attention Scores and citations themselves, fold-changes of the two metrics were weakly correlated with each other (Spearman correlation 0.19).

To quantify the overall evidence for each variable's association with Attention Score and citations, we performed a random-effects meta-analysis of the respective model coefficients (*Table 1* and *Supplementary file 8*). Based on the meta-analysis, an article's Attention Score and citations were positively associated with its preprint status, number of authors, number of

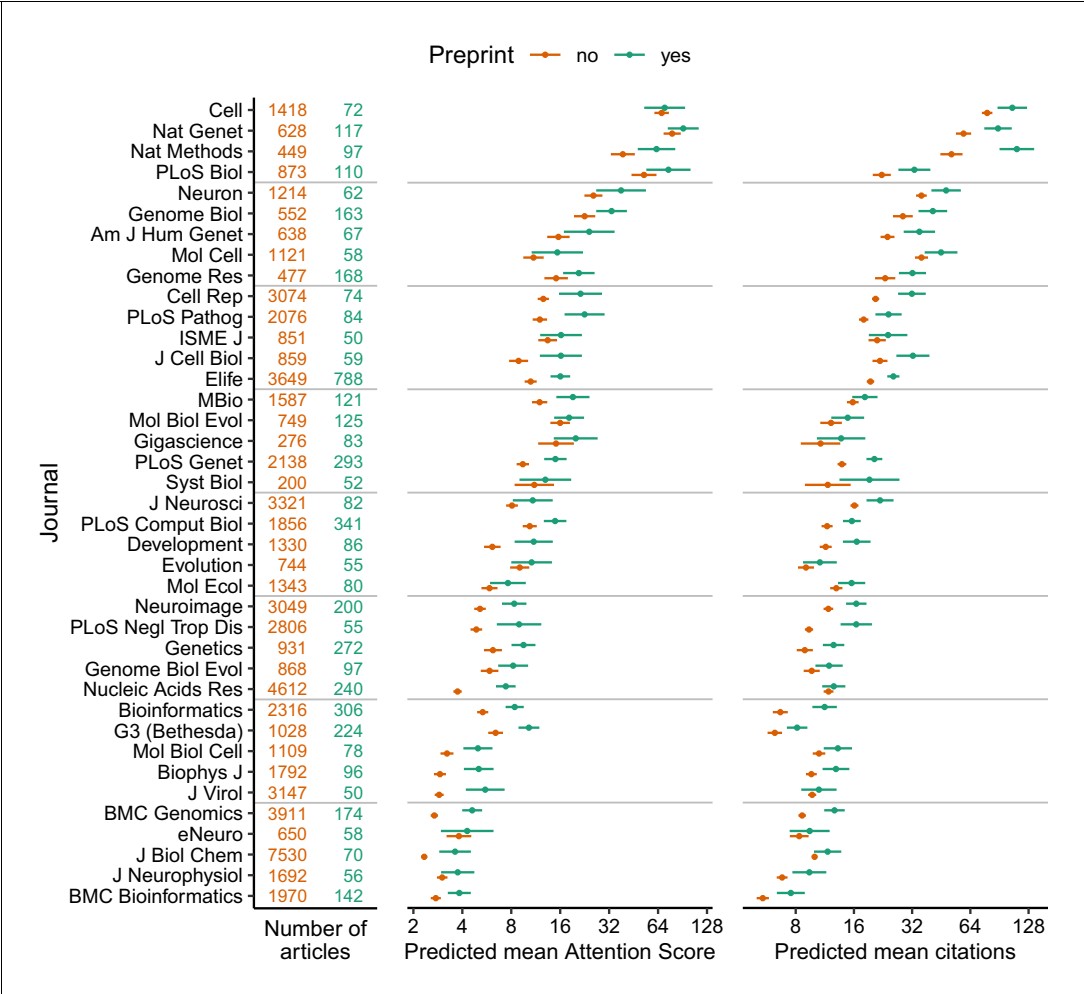

**Figure 1.** Absolute effect size of having a preprint, by metric (Attention Score and number of citations) and journal. Each point indicates the predicted mean of the Attention Score (middle column) and number of citations (right column) for a hypothetical article with (green) or without (orange) a preprint, assuming the hypothetical article was published three years ago and had the mean value (i.e., zero) of each of the top 15 MeSH term PCs and the median value (for articles in that journal) of number of authors, number of references, U.S. affiliation status, Nature Index affiliation status, and last author publication age. Error bars indicate 95% confidence intervals. Journal names correspond to PubMed abbreviations: number of articles with (green) and without (orange) a preprint are shown in the left column. Journals are ordered by the mean of predicted mean Attention Score and predicted mean number of citations.

The online version of this article includes the following source data and figure supplement(s) for figure 1:

**Source data 1.** Absolute effect size of having a preprint, by metric and journal.
**Figure supplement 1.** Accuracy of automatically inferring last-author publications from names and affiliations in PubMed.
**Figure supplement 2.** Histogram of the number of days by which release of the preprint preceded publication of the peer-reviewed article, including articles from all journals.
**Figure supplement 3.** Scatterplots of Attention Score (with a pseudocount of 1) for articles in each journal.
**Figure supplement 4.** Scatterplots of number of citations (with a pseudocount of 1) for articles in each journal.
**Figure supplement 5.** Scatterplots of number of citations vs. Attention Score for articles in each journal.
**Figure supplement 6.** Percentage of variance in MeSH term assignment explained by the top 15 principal components for each journal.
**Figure supplement 7.** Scores for the top two principal components of MeSH term assignments for each journal.
**Figure supplement 8.** Comparing mean absolute error (MAE) and mean absolute percentage error (MAPE) of Gamma and log-linear regression models for each metric.
**Figure supplement 9.** Absolute effect size of having a preprint, by metric and journal.

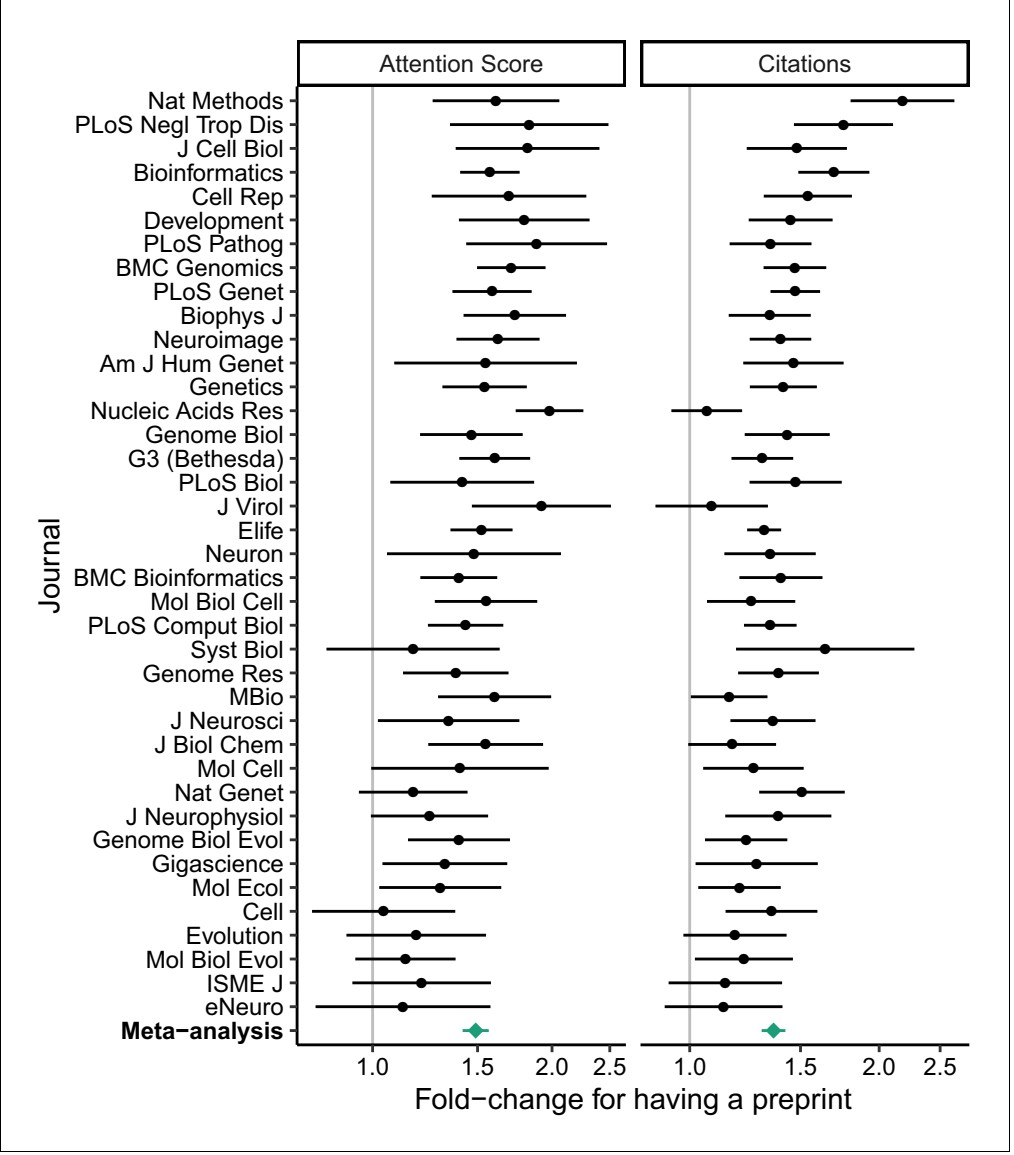

**Figure 2.** Relative effect size of having a preprint, by metric (Attention Score and number of citations) and journal. Fold-change corresponds to the exponentiated coefficient from log-linear regression, where fold-change >1 indicates higher Attention Score or number of citations for articles that had a preprint. A fold-change of 1 corresponds to no association. Error bars indicate 95% confidence intervals. Journals are ordered by mean log fold-change. Bottom row shows estimates from random-effects meta-analysis (also shown in *Table 1*). The source data for this figure is in *Supplementary file 7*.

The online version of this article includes the following figure supplement(s) for figure 2:

**Figure supplement 1.** Associations of MeSH term PCs with Attention Score and citations in each journal, based on model coefficients from log-linear regression.

**Figure supplement 2.** Comparing model fits with and without MeSH term PCs.

references, U.S. affiliation status, and Nature Index affiliation status, and slightly negatively associated with its last author publication age.

In particular, having a preprint was associated with a 1.49 times higher Attention Score (95% CI 1.42 to 1.57) and 1.36 times more citations (95% CI 1.30 to 1.42) of the peer-reviewed article.

These effect sizes were ~4 times larger than those for having an author with an affiliation in the U.S. or at a Nature Index institution. In a separate meta-analysis, the amount of time between release of the preprint and publication of the article was positively associated with the article's Attention Score, but not its citations

**Table 1.** Random-effects meta-analysis across journals of model coefficients from log-linear regression.
A positive coefficient (column 3) means that Attention Score or number of citations increases as that variable increases (or if the article had a preprint or had an author with a U.S. affiliation or a Nature Index affiliation). However, coefficients for some variables have different units and are not directly comparable. P-values were adjusted using the Bonferroni-Holm procedure, based on having fit two models for each journal. Effectively, for each variable, the procedure multiplied the lesser p-value by two and left the other unchanged. Meta-analysis statistics for the intercept and publication date are shown in *Supplementary file 8*.

| Metric | Article-level variable | Coef. | Std. error | 95% CI (lower) | 95% CI (upper) | p-value | Adj. p-value |
|---|---|---|---|---|---|---|---|
| Attention Score | Had a preprint | 0.575 | 0.036 | 0.502 | 0.647 | 1.91e-18 | 3.82e-18 |
| | $\log_2$(number of authors) | 0.129 | 0.015 | 0.099 | 0.158 | 1.04e-10 | 1.04e-10 |
| | $\log_2$(number of references + 1) | 0.070 | 0.021 | 0.027 | 0.113 | 2.10e-03 | 2.10e-03 |
| | Had an author with U.S. affiliation | 0.143 | 0.021 | 0.100 | 0.187 | 6.08e-08 | 6.08e-08 |
| | Had an author with Nature Index affiliation | 0.147 | 0.020 | 0.106 | 0.188 | 1.20e-08 | 2.41e-08 |
| | Last author publication age (yrs) | −0.009 | 0.001 | −0.011 | −0.007 | 5.86e-10 | 1.17e-09 |
| Citations | Had a preprint | 0.442 | 0.031 | 0.380 | 0.505 | 7.38e-17 | 7.38e-17 |
| | $\log_2$(number of authors) | 0.181 | 0.009 | 0.163 | 0.200 | 9.76e-22 | 1.95e-21 |
| | $\log_2$(number of references + 1) | 0.217 | 0.020 | 0.176 | 0.258 | 4.87e-13 | 9.73e-13 |
| | Had an author with U.S. affiliation | 0.079 | 0.011 | 0.057 | 0.102 | 1.49e-08 | 2.98e-08 |
| | Had an author with Nature Index affiliation | 0.100 | 0.015 | 0.071 | 0.130 | 3.46e-08 | 3.46e-08 |
| | Last author publication age (yrs) | −0.003 | 0.001 | −0.004 | −0.001 | 8.61e-05 | 8.61e-05 |

(*Supplementary files 9* and *10*). Taken together, these results indicate that having a preprint is associated with a higher Attention Score and more citations independently of other article-level variables.

Because we calculated the MeSH term PCs separately for each journal, we did not perform a random-effects meta-analysis of the corresponding coefficients. However, within each journal, typically several PCs had p-value≤0.05 for association with Attention Score or citations (*Figure 2—figure supplement 1*). In addition, if we excluded the MeSH term PCs from the regression, the fold-changes for having a preprint increased modestly (*Figure 2—figure supplement 2* and *Supplementary file 11*). These results suggest that the MeSH term PCs capture meaningful variation in scientific subfield between articles in a given journal.

Finally, using meta-regression, we found that the log fold-changes of the two metrics were not associated with the journal's access model, Impact Factor, or percentage of articles with preprints (*Table 2* and *Supplementary file 12*). Thus, these journal-level characteristics do not explain journal-to-journal variation in the differences in Attention Score and citations between articles with and without a preprint.

## Discussion

The decision of when and where to disclose the products of one's research is influenced by multiple factors. Here we find that having a preprint on bioRxiv is associated with a higher Altmetric Attention Score and more citations of the peer-reviewed article. The associations appear independent of several other article- and author-level variables and unrelated to journal-level variables such as access model and Impact Factor.

The advantage of stratifying by journal as we did here is that it accounts for the journal-specific factors – both known and unknown – that affect an article's Attention Score and citations. The disadvantage is that our results only apply to journals that have published at least 50 articles that have a preprint on bioRxiv (with multidisciplinary journals excluded). In fact, our preprint counts may be an underestimate, since some preprints on bioRxiv have been published as peer-reviewed articles, but not yet detected as such by bioRxiv's internal system (*Abdill and Blekhman, 2019*). Furthermore, the associations we observe may not apply to preprints on other repositories such as arXiv Quantitative Biology and PeerJ Preprints.

We used the Altmetric Attention Score and number of citations on CrossRef because, unlike other article-level metrics such as number of views, both are publicly and programmatically available for any article with a DOI. However, both metrics are only crude proxies for an

**Table 2.** Meta-regression across journals of log fold-changes for having a preprint.

A positive coefficient means the log fold-change for having a preprint increases as that variable increases (or if articles in that journal are immediately open access). However, coefficients for different variables have different units and are not directly comparable. P-values were adjusted using the Bonferroni-Holm procedure, based on having fit two models. Depending on the two p-values for a given variable, the procedure may have left one p-value unchanged. Regression statistics for the intercept are shown in *Supplementary file 12*.

| Metric | Journal-level variable | Coef. | Std. error | 95% CI (lower) | 95% CI (upper) | t-statistic | p-value | Adj. p-value |
|---|---|---|---|---|---|---|---|---|
| Attention Score | Immediately open access | 0.118 | 0.076 | −0.037 | 0.273 | 1.551 | 0.130 | 0.260 |
| | $\log_2$(Impact Factor) | −0.025 | 0.040 | −0.107 | 0.057 | −0.616 | 0.542 | 0.542 |
| | $\log_2$(% of articles with preprints) | −0.064 | 0.032 | −0.129 | 0.001 | −1.991 | 0.054 | 0.109 |
| Citations | Immediately open access | −0.013 | 0.069 | −0.152 | 0.126 | −0.187 | 0.853 | 0.853 |
| | $\log_2$(Impact Factor) | 0.044 | 0.036 | −0.030 | 0.117 | 1.211 | 0.234 | 0.468 |
| | $\log_2$(% of articles with preprints) | 0.037 | 0.029 | −0.022 | 0.095 | 1.283 | 0.208 | 0.208 |

article's true scientific impact, which is difficult to quantify and can take years or decades to assess.

For multiple reasons, our analysis does not indicate whether the associations between preprints, Attention Scores, and citations have changed over time. First, historical citation counts are not currently available from CrossRef, so our data included each article's citations at only one moment in time. Second, most journals had a relatively small number of articles with preprints and most preprints were relatively recent, so we did not model a statistical interaction between publication date and preprint status. We also largely ignored characteristics of the preprints themselves. In any case, the associations we observe may change as the culture of preprints in the life sciences evolves.

Grouping scientific articles by their research areas is an ongoing challenge (*Piwowar et al., 2018*; *Waltman and van Eck, 2012*). Although the principal components of MeSH term assignments are only a simple approximation, they do explain some variation in Attention Score and citations between articles in a given journal. Thus, our approach to estimating scientific subfield may be useful in other analyses of the biomedical literature.

Our heuristic approach to infer authors' publication histories from their names and free-text affiliations in PubMed was accurate, but not perfect. The heuristic was necessary because unique author identifiers such as ORCID iDs currently have sparse coverage of the published literature. This may change with a recent requirement from multiple U.S. funding agencies (*NIH, 2019*), which would enhance future analyses of scientific publishing.

Because our data are observational, we cannot conclude that releasing a preprint is causal for a higher Attention Score and more citations of the peer-reviewed article. Even accounting for all the other factors we modeled, having a preprint on bioRxiv could be merely a marker for research likely to receive more attention and citations anyway. For example, perhaps authors who release their work as preprints are more active on social media, which could partly explain the association with Attention Score, although given the weak correlation between Attention Score and citations, it would likely not explain the association with citations. If there is a causal role for preprints, it may be related to increased visibility that leads to "preferential attachment" (*Wang et al., 2013*) while the manuscript is in peer review. These scenarios need not be mutually exclusive, and without a randomized trial they are extremely difficult to distinguish.

Altogether, our findings contribute to the growing observational evidence of the effects of preprints in biology (*Fraser et al., 2019*), and have implications for preprints in chemistry and medicine (*Kiessling et al., 2016*; *Rawlinson and Bloom, 2019*). Consequently, our study may help researchers and publishers make informed decisions about how to incorporate preprints into their work.

## Methods

### Collecting the data

Data came from four primary sources: PubMed, Altmetric, CrossRef, and Rxivist. We obtained data for peer-reviewed articles from PubMed using NCBI's E-utilities API via the rentrez R package (*Winter, 2017*). We obtained Altmetric

Attention Scores using the Altmetric Details Page API via the rAltmetric R package. The Altmetric Attention Score is an aggregate measure of mentions from various sources, including social media, mainstream media, and policy documents (https://www.altmetric.com/about-our-data/our-sources/). We obtained numbers of citations using the CrossRef API (specifically, we used "is-referenced-by-count"). We obtained links between bioRxiv preprints and peer-reviewed articles using the CrossRef API via the rcrossref R package. We verified and supplemented the links from CrossRef using Rxivist (*Abdill and Blekhman, 2019*) via the Postgres database in the public Docker image (https://hub.docker.com/r/blekhmanlab/rxivist_data). We merged data from the various sources using the Digital Object Identifier (DOI) and PubMed ID of the peer-reviewed article.

We obtained Journal Impact Factors from the 2018 Journal Citation Reports published by Clarivate Analytics. We obtained journal access models from the journals' websites. As in previous work (*Abdill and Blekhman, 2019*), we classified access models as "immediately open" (in which all articles receive an open access license immediately upon publication) or "closed or hybrid" (anything else).

Starting with all publications indexed in PubMed, we applied the following inclusion criteria:

- Published between January 1, 2015 and December 31, 2018 (inclusive). Since bioRxiv began accepting preprints on November 7, 2013, our start date ensured sufficient time for the earliest preprints to be published.
- Had a DOI. This was required for obtaining Attention Score and number of citations, and excluded many commentaries and news articles.
- Had a publication type in PubMed of Journal Article and not Review, Published, Erratum, Comment, Lecture, Personal Narrative, Retracted Publication, Retraction of Publication, Biography, Portrait, Autobiography, Expression of Concern, Address, or Introductory Journal Article. This filtered for original research articles.
- Had at least one author. A number of editorials met all of the above criteria, but lacked any authors.
- Had an abstract of sufficient length. A number of commentaries and news articles met all of the above criteria, but either lacked an abstract or had an anomalously short one. We manually inspected articles

with short abstracts to determine a cutoff for each journal (*Supplementary file 3*).
- Had at least one Medical Subject Headings (MeSH) term. Although not all articles from all journals had MeSH terms (which are added by PubMed curators), this requirement allowed us to adjust for scientific subfield within a journal using principal components of MeSH terms.

Inclusion criteria for bioRxiv preprints:

- Indexed in CrossRef or Rxivist as linked to a peer-reviewed article in our dataset.
- Released prior to publication of the corresponding peer-reviewed article.

Inclusion criteria for journals:

- Had at least 50 peer-reviewed articles in our dataset previously released as preprints. Since we stratified our analysis by journal, this requirement ensured a sufficient number of peer-reviewed articles to reliably estimate each journal's model coefficients and confidence intervals (*Austin and Steyerberg, 2015*).
- We excluded the multidisciplinary journals Nature, Nature Communications, PLoS One, PNAS, Royal Society Open Science, Science, Science Advances, and Scientific Reports, since some articles published by these journals would likely not be released on bioRxiv, which could have confounded the analysis.

We obtained all data on September 28, 2019, thus all predictions of Attention Score and citations are for this date. Preprints and peer-reviewed articles have distinct DOIs, and accumulate Attention Scores and citations independently of each other. We manually inspected 100 randomly selected articles from the final set, and found that all 100 were original research articles. For those 100 articles, the Spearman correlation between number of citations from CrossRef and number of citations from Web of Science Core Collection was 0.98, with a mean difference of 2.5 (CrossRef typically being higher), which indicates that the citation data from CrossRef are reliable and different sources would likely not produce different results.

### Inferring author-level variables

Institutional affiliation in PubMed is a free-text field, but is typically a series of comma-separated values with the country near the end. To identify the corresponding country of each affiliation, we used a series of heuristic regular

expressions (*Supplementary file 13* shows the number of affiliations for each identified country). Each author of a given article can have zero or more affiliations. For many articles, especially less recent ones, only the first author has any affiliations listed in PubMed, even though those affiliations actually apply to all the article's authors (as verified by the version on the journal's website). Therefore, the regression modeling used a binary variable for each article corresponding to whether any author had any affiliation in the United States.

To approximate institutions that may be associated with higher citation rates, we used the 2019 Nature Index for Life Sciences (*Nature Index, 2019*), which lists the 100 institutions with the highest fractional count of articles in Nature Index journals in the Life Sciences between January 1, 2018 and December 31, 2018. The fractional count accounts for the fraction of authors from that institution and the number of affiliated institutions per article. Nature Index journals are selected by panels of active scientists and are supposed to represent the "upper echelon" (*Nature Index, 2014*). They are not limited to journals of Nature Publishing Group. We used regular expressions to identify which affiliations corresponded to which Nature Index institutions. The regression modeling then used a binary variable for each article corresponding to whether any author had an affiliation at any of the Nature Index institutions.

For each article in our dataset, we sought to identify the last author's *first* last-author publication, i.e., the earliest publication in which that person is the last author, in order to estimate how long a person has been a principal investigator. Author disambiguation is challenging, and unique identifiers are currently sparse in PubMed and bioRxiv. We developed an approach to infer an author's previous publications in PubMed based only on that person's name and affiliations.

The primary components of an author's name in PubMed are last name, fore name (which often includes middle initials), and initials (which do not include last name). Fore names are present in PubMed mostly from 2002 onward. For each article in our dataset (each target publication), our approach went as follows:

1. Get the last author's affiliations for the target publication. If the last author had no direct affiliations, get the affiliations of the first author. These are the target affiliations.

2. Find all publications between January 1, 2002 and December 31, 2018 in which the last author had a matching last name and fore name. We limited the search to last-author publications to approximate publications as principal investigator and to limit computation time. These are the query publications.

3. For each query publication, get that author's affiliations. If the author had no direct affiliations, get the affiliations of the first author. These are the query affiliations.

4. Clean the raw text of all target and query affiliations (make all characters lowercase and remove non-alphanumeric characters, among other things).

5. Calculate the similarity between each target-affiliation-query-affiliation pair. Similarity was a weighted sum of the shared terms between the two affiliations. Term weights were calculated using the quanteda R package (*Benoit et al., 2018*) and based on inverse document frequency, i.e., $\log_{10}(1/\text{frequency})$, from all affiliations from all target publications in our dataset. Highly common (frequency >0.05), highly rare (frequency $<10^{-4}$), and single-character terms were given no weight.

6. Find the earliest query publication for which the similarity between a target affiliation and a query affiliation is at least 4. This cutoff was manually tuned.

7. If the earliest query publication is within two years of when PubMed started including fore names, repeat the procedure using last name and initials instead of last name and fore name.

For a randomly selected subset of 50 articles (none of which had been used to manually tune the similarity cutoff), we searched PubMed and authors' websites to manually identify each last author's first last-author publication. The Spearman correlation between manually identified and automatically identified dates was 0.88, the mean error was 1.74 years (meaning our automated approach sometimes missed the earliest publication), and the mean absolute error was 1.81 years (*Figure 1—figure supplement 1*). The most common reason for error was that the author had changed institutions (*Supplementary file 14*).

## Calculating principal components of MeSH term assignments

Medical Subject Headings (MeSH) are a controlled vocabulary used to index PubMed and other biomedical databases. For each journal,

we generated a binary matrix of MeSH term assignments for the peer-reviewed articles (1 if a given term was assigned to a given article, and 0 otherwise). We only included MeSH terms assigned to at least 5% of articles in a given journal, and excluded the terms "Female" and "Male" (which referred to the biological sex of the study animals and were not related to the article's field of research). We calculated the principal components (PCs) using the prcomp function in the stats R package and scaling the assignments for each term to have unit variance. We calculated the percentage of variance in MeSH term assignment explained by each PC as that PC's eigenvalue divided by the sum of all eigenvalues. By calculating the PCs separately for each journal, we sought to capture the finer variation between articles in a given journal rather than the higher-level variation between articles in different journals.

### Quantifying the associations

Attention Scores are real numbers $\geq 0$, whereas citations are integers $\geq 0$. Therefore, for each journal, we fit two types of regression models for Attention Score and three for citations:

- Log-linear regression, in which the dependent variable was $\log_2$(Attention Score + 1) or $\log_2$(citations + 1).
- Gamma regression with a log link, in which the dependent variable was "Attention Score + 1" or "citations + 1". The response variable for Gamma regression must be >0.
- Negative binomial regression, in which the dependent variable was citations. The response variable for negative binomial regression must be integers $\geq 0$.

Each model had the following independent variables for each peer-reviewed article:

- Preprint status, encoded as 1 for articles preceded by a preprint and 0 otherwise.
- Publication date (equivalent to time since publication), encoded using a natural cubic spline with three degrees of freedom. The spline provides flexibility to fit the non-linear relationship between citations (or Attention Score) and publication date. In contrast to a single linear term, the spline does not assume, for example, that the average difference in the dependent variable between a 0-year-old article and a 1-year-old article is the same as between a 4-year-old article and a 5-year-old article. Source: PubMed.

- Number of authors, log-transformed because it was strongly right-skewed. Source: PubMed.
- Number of references, log-transformed because it was strongly right-skewed. Sources: PubMed and CrossRef. For some articles, either PubMed or CrossRef lacked complete information on the number of references. For each article, we used the maximum between the two.
- U.S. affiliation status, encoded as 1 for articles for which any author had a U.S. affiliation and 0 otherwise. Source: inferred from PubMed as described above.
- Nature Index affiliation status, encoded as 1 for articles for which any author had an affiliation at an institution in the 2019 Nature Index for Life Sciences and 0 otherwise. Source: inferred from PubMed and the Nature Index data as described above.
- Last author publication age, encoded as the amount of time in years by which publication of the peer-reviewed article was preceded by publication of the last author's *first* last-author publication. Source: inferred from PubMed as described above.
- Top 15 PCs of MeSH term assignments (or all PCs, if there were fewer than 15). Source: calculated from PubMed as described above. Calculating the MeSH term PCs and fitting the regression models on a journal-wise basis means, for example, that the effect on Attention Score and citations of publishing a paper about *Saccharomyces cerevisiae* or about diffusion magnetic resonance imaging depends on whether the paper is in Molecular Cell or in Neuroimage.

We evaluated goodness-of-fit of each regression model using mean absolute error and mean absolute percentage error. To fairly compare the different model types, we converted each prediction to the original scale of the respective metric prior to calculating the error.

As a secondary analysis, we added to the log-linear regression model a variable corresponding to the amount of time in years by which release of the preprint preceded publication of the peer-reviewed article (using 0 for articles without a preprint). We calculated this variable based on preprint release dates from CrossRef and Rxivist and publication dates from PubMed.

We extracted coefficients and their 95% confidence intervals from each log-linear regression model. Because preprint status is binary, its model coefficient corresponded to a $\log_2$ fold-change. We used each regression model to

calculate predicted Attention Score and number of citations, along with corresponding 95% confidence intervals and 95% prediction intervals, given certain values of the variables in the model. For simplicity in the rest of the manuscript, we refer to exponentiated model coefficients as fold-changes of Attention Score and citations, even though they are actually fold-changes of "Attention Score + 1" and "citations + 1".

We performed each random-effects meta-analysis based on the Hartung-Knapp-Sidik-Jonkman method (*IntHout et al., 2014*) using the metagen function of the meta R package (*Schwarzer et al., 2015*). We performed meta-regression by fitting a linear regression model in which the dependent variable was the journal's coefficient for preprint status (from either Attention Score or citations) and the independent variables were the journal's access model (encoded as 0 for "closed or hybrid" and 1 for "immediately open"), $\log_2$(Impact Factor), and $\log_2$(percentage of articles released as preprints). We adjusted p-values for multiple testing using the Bonferroni-Holm procedure, which is uniformly more powerful than the standard Bonferroni procedure (*Holm, 1979*).

### Data availability

Code and data to reproduce this study are available on Figshare (https://doi.org/10.6084/m9.figshare.8855795).

### Acknowledgements

We thank Altmetric for providing their data free of charge for research purposes. We thank Tony Capra and Doug Ruderfer for helpful comments on the manuscript.

**Darwin Y Fu** is in the Department of Biomedical Informatics, Vanderbilt University Medical Center, Nashville, United States
🆔 https://orcid.org/0000-0003-1407-1689

**Jacob J Hughey** is in the Department of Biomedical Informatics and the Department of Biological Sciences, Vanderbilt University Medical Center, Nashville, United States
jakejhughey@gmail.com
🆔 https://orcid.org/0000-0002-1558-6089

*Author contributions:* Darwin Y Fu, Investigation, Visualization, Methodology, Writing - review and editing; Jacob J Hughey, Conceptualization, Supervision, Funding acquisition, Investigation, Visualization, Methodology, Writing - original draft, Writing - review and editing

*Competing interests:* The authors declare that no competing interests exist.

### Funding

| Funder | Grant reference number | Author |
|---|---|---|
| National Institute of General Medical Sciences | R35GM124685 | Jacob J Hughey |
| U.S. National Library of Medicine | T15LM007450 | Darwin Y Fu |

The funders had no role in study design, data collection and interpretation, or the decision to submit the work for publication.

### Decision letter and Author response

Decision letter https://doi.org/10.7554/eLife.52646.sa1
Author response https://doi.org/10.7554/eLife.52646.sa2

## Additional files

### Supplementary files

• Supplementary file 1. Characteristics of journals included in this study.

• Supplementary file 2. Spearman correlation between Attention Score and number of citations for articles in each journal.

• Supplementary file 3. Journal-specific cutoffs of minimum abstract length for including articles in our dataset. Articles from journals not specified in this table were included regardless of abstract length.

• Supplementary file 4. Descriptive statistics for each variable included in the regression models for each journal. For logical variables, an entry "logical | $n_t$ | $n_f$" corresponds to $n_t$ articles for which the variable was true and $n_f$ articles for which the variable was false. For numeric variables and dates, each entry corresponds to "minimum | first quartile | median | mean | third quartile | maximum".

• Supplementary file 5. MeSH terms with the highest positive and negative loadings for each PC and each journal.

• Supplementary file 6. Mean absolute error (MAE) and mean absolute percentage error (MAPE) of log-linear regression, gamma regression, and negative binomial regression for each metric and each journal.

• Supplementary file 7. Regression statistics from log-linear regression for each metric, journal, and variable.

• Supplementary file 8. Random-effects meta-analysis across journals for each metric and variable, based on the regression statistics in *Supplementary file 7*.

• Supplementary file 9. Regression statistics from log-linear regression for each metric, journal, and variable, including as a variable the amount of time in years between release of the preprint and publication of the peer-reviewed article ("preprint_age").

• Supplementary file 10. Random-effects meta-analysis across journals for each metric and variable, based on the regression statistics in *Supplementary file 9*.

• Supplementary file 11. Regression statistics from log-linear regression for each metric, journal, and variable, excluding as variables the MeSH term PCs.

• Supplementary file 12. Meta-regression statistics of log fold-changes for having a preprint.

• Supplementary file 13. Frequencies of country of affiliation inferred using free-text affiliations from PubMed.

• Supplementary file 14. Comparison of automatically identified and manually identified earliest last-author publications for 50 randomly selected articles.

• Transparent reporting form

### Data availability

Code and data to reproduce this study are available on Figshare (https://doi.org/10.6084/m9.figshare.8855795). In accordance with Altmetric's data use agreement, the Figshare repository does not include each article's Altmetric data, which are available from Altmetric after obtaining an API key.

The following dataset was generated:

| Author(s) | Year | Dataset URL | Database and Identifier |
|---|---|---|---|
| Jacob J Hughey | 2019 | https://doi.org/10.6084/m9.figshare.8855795 | figshare, 10.6084/m9.figshare.8855795 |

The following previously published dataset was used:

| Author(s) | Year | Dataset URL | Database and Identifier |
|---|---|---|---|
| Abdill RJ, Blekhman R | 2019 | https://hub.docker.com/r/blekhmanlab/rxivist_data | Docker Hub, blekhmanlab/rxivist_data:2019-08-30 |

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
