## [Decision Letter]

[Editors’ note: the original version of this study was declined after peer review. The authors submitted a revised version, which was reviewed, revised again, and accepted. The decision letter sent after the first round of peer review is immediately below, followed by the decision letter sent after the second round of peer review.]

Thank you for submitting your manuscript "Releasing a preprint is associated with more attention and citations" to *eLife* for consideration as a Feature Article. Your article has been reviewed by three peer reviewers, and the evaluation has been overseen by the *eLife* Features Editor.

Our decision has been reached after consultation between the reviewers. Based on these discussions and the individual reviews below, we regret to inform you that your work will not be considered further for publication in *eLife*.

Reviewer #1:

The submitted manuscript, "Releasing a preprint is associated with more attention and citations," examines more than 45,000 publications from 26 biology journals and compares the citation counts and Altmetric Attention Scores of publications that were or were not previously released as a preprint. After controlling for journal, publication date and a cleverly constructed "scientific subfield," the authors find articles that had been pre-printed are associated with higher citation counts and Attention Scores, and that these effects are diminished in journals with higher Journal Impact Factors. The code availability is exemplary, the results are conveyed clearly, and it's laudable that the Discussion section is so frank regarding the limitations of the study. However, I am unsure whether those limitations are acceptable – if they are, there are several issues that should be addressed before publication.

My primary concern is the large number of confounders that are unaccounted for in the study. Though the authors make a convincing case that the publications with preprints do have an advantage in citations and attention score, there is little attention given in the analysis to the numerous factors that have already been linked to increased attention or citations, factors that may be the actual driver of this effect, rendering preprint status irrelevant. For example, perhaps (published) preprints are longer than articles without preprints [1,2] or have more authors [2]. The results could be affected if senior authors with other highly cited papers are more likely to post a preprint [3,4], or if bioRxiv has an overrepresentation of researchers from "elite" institutions [5] or large research groups [6]. Or maybe authors enthusiastic about preprints also just happen to be more active on Twitter.

In short, I am unsure of the utility of measuring complex dynamics like citation count and attention score without incorporating any data about authors. Most importantly, I do not believe this work provides adequate support for the statement in its Abstract that "this observational study can help researchers and publishers make informed decisions about how to incorporate preprints into their work." This assertion is directly contradicted by the Discussion section, which accurately assesses the shortcomings of the study. I sympathize with the difficulties in obtaining data on authors, and I do not believe those shortcomings could be remediated in the two-month revision timeline given in the review instructions. I leave it to the discretion of the editors whether this represents an incremental improvement significant enough for publication.

Additional comments:

Results, first paragraph: I believe there are several ways for this finding to be strengthened, some of which pose significant problems as presented currently:

- Most straightforwardly, I don't believe the statement accurately reflects what is actually being measured: "Across journals, each article's Attention Score and citations were weakly correlated" does not convey that the metrics within each journal, and then again within individual years, are, on average, weakly correlated. Clarification would be helpful.

- This result would also benefit from elaboration, here or in the Materials and methods section, regarding why the data were segmented this way to begin with. While other results indicate that journal and year both influence the *difference* between preprinted and non-preprinted publications, is there any evidence that journal and/or year influence the relationship between citations and attention score overall? Comparing all articles together, it appears citations and attention score are correlated with Spearman's rho=0.32. Is there any benefit to instead separating the articles into "journal-years," calculating separate rho values, and finding the median is instead 0.29? If it is to measure the consistency of the relationship across years and journals, this should be stated more explicitly.

- The reported median Spearman correlation and the contents of Figure S2 exclude any consideration of the significance of the correlations, which, while it may not substantially affect the median, does obscure how frequently a journal-year does not have a significant correlation between citations and attention score. A quick estimation using the p-values from the "cor.test" function, corrected for performing 103 correlation tests, suggests there are 16 journal-years that have no significant correlation-changing those rho values to 0 makes Figure S2 look very different, and the largest bin becomes the one indicating no correlation at all. That said, I don't have the knowledge to say if the assumptions used to generate p-values in that package are appropriate here, since the end result is a median of correlations. At the least, if the authors keep the median Spearman correlation, this finding would benefit from performing a permutation test to help us understand whether this median is any better than would be expected at random: Scrambling the citation numbers (possibly within each journal?) and calculating a new median, say, 1,000 times, may help reveal how unexpected a median rho of 0.29 actually is.

Results, second paragraph: I'm curious about the reasoning given for using a log transformation here: "since both metrics were greater than or equal to zero and spanned orders of magnitude." While those are reasonable tests to see whether it's *possible* to log-transform the data, it seems like the actual reason was because it enables using the regression to discuss log-fold change rather than differences in absolute measurements. This statement would benefit from clarification, or a citation supporting why the stated criteria is appropriate.

Table 2: The authors state that they present uncorrected p-values here because "for each metric, the three variables were tested in one model." This is true, and a nice benefit of meta-regression. However, the paper describes the results of two *different* meta-regression models, which test two different hypotheses (one regarding attention score, and another regarding citations). Though a p-value threshold is not specified for these tests (Results, final paragraph, states only "significantly associated"), convention would suggest the cutoff was 0.05. A Bonferroni correction (for two tests) would push all p values in Table 2 above this threshold-the authors should correct these measurements and update the findings described in the final paragraph of the Results accordingly. Alternatively, it would be acceptable to leave these values as-is and provide further support for not correcting the p-values generated by multiple meta-regression analyses.

References:

(Not intended as a request for the authors to include citations to these papers, just offered here to support my feedback above.)

1) Falagas et al. 2013. The Impact of Article Length on the Number of Future Citations: A Bibliometric Analysis of General Medicine Journals. PLOS ONE. doi: 10.1371/journal.pone.0049476.

2) Fox et al. 2016. Citations increase with manuscript length, author number, and references cited in ecology journals. Ecology and Evolution. doi: 10.1002/ece3.2505.

3) Fu and Aliferis. 2008. Models for Predicting and Explaining Citation Count of Biomedical Articles. AMIA Annual Symposium Proceedings. PMID: 8999029.

4) Perc. 2014. The Matthew effect in empirical data. Journal of the Royal Society Interface. doi: 10.1098/rsif.2014.0378.

5) Medoff. 2007. Evidence of a Harvard and Chicago Matthew Effect. Journal of Economic Methodology. doi: 10.1080/13501780601049079.

6) van Raan. 2006. Performance‐related differences of bibliometric statistical properties of research groups: Cumulative advantages and hierarchically layered networks. Journal of the American Society for Information Science and Technology. doi: 10.1002/asi.20389.

Reviewer #2:

The article investigates the relationship between preprint posting and subsequent citations and Altmetric scores of published articles. The topic is interesting and worthy of attention and, although it has been approached previously in the literature with similar conclusions (Serghiou and Ioannidis, 2018), the authors perform a much more detailed analysis of the subject. Data and code for the performed analyses are available in figshare.

My major concerns and suggestions are the following:

1) Although the statistics used, based on multivariate linear regression, are generally solid, the visualization of results is frequently less intuitive than it could be, and seems to be aimed at the data scientist more than at the average reader. In summary:

- In Figure 1A, although the forest plot is a great way to show how results vary by journal, it does not allow the reader to appreciate the range of variability of citations among articles with and without preprints (as the confidence intervals shown depend both on variability and on sample size). As the article is ultimately the unit of analysis here, I miss a standard scatter plot showing unit-level citations for articles with and without preprints – either for the whole set of articles or divided by journal (although this might be too large and better left to a supplementary figure).

- Again in terms of visualizing variability, are the authors sure about using log scales in the figures? Although logarithmic transformation is useful for statistics, it is once again not the most intuitive way for the reader to get a sense of the full range of variability.

- Some of the supplementary figures (particularly Figures S3 and S5) are quite hard to read and understand due to the sheer volume of data presented and might not be very useful to the reader.

2) Did the author evaluate the interaction between preprints and publication year in the model? This seems like an important question to better assess the likelihood of causality between preprint availability and citations – for more recent articles, I would expect the preprint-related advantage to be greater, as the preprint-only period will account for a larger part of the paper's lifetime. Over time, this advantage is likely to taper off. On the other hand, if an interaction between publication year and preprint availability is not observed, this might suggest that the citation advantage is not necessarily due to the preprint's visibility, but rather to other features or confounders (e.g. articles with preprints being of higher interest, or stemming from groups with more visibility).

3) Similarly, it would be interesting to investigate whether metrics of the preprint such as altmetric scores, number of downloads, or even citations – all of which are available in bioRxiv or CrossRef) are related to the citation advantage for their published versions. Although showing that the advantage is larger for preprints with more visibility does not prove that posting a preprint leads to more citations, finding that the advantage does not correlate with attention received to the preprint would argue against it. All of these could be performed as secondary analyses similar to the one performed for the number of days the preprint preceded the peer-reviewed article.

4) The criteria for journal selection should be explained more clearly for the reader to assess the sample’s representativeness. Moreover, the principal components used in subfield analysis could be better described (explicitly showing the terms with high loading in each of them) for one to get a sense of how meaningful they are to define subfields. Both of these themes are expanded in the minor comments.

Reviewer #3:

This article first identified 26 journals in the life sciences that have published at least 50 articles with a preprint on bioRxiv. For each article published in these journals between 2015-2018 it then extracted citation count from CrossRef, attention score from Altmetric and presence of a preprint from CrossRef and Rxivist. It then used log-transformed linear regression to quantify the association of having a preprint to citation count and attention score, adjusting for time since publication using a spline and scientific field using principal components analysis (PCA) of MeSH terms. It finally used meta-regression to conclude that across journals, overall, having a preprint was associated with 1.53 higher attention score and 1.31 higher citation count.

The authors should be commended for attempting to study and promote the advantages of preprints, validate and extend previous work, for openly providing their data and code and for including a good limitations section. However, the chosen method of data collection is highly prone to selection bias, many of the descriptive and analytic choices are poorly justified and the magnitude of association, wherever presented, is quantified in relative rather than absolute terms, which in the setting of highly skewed data is very misleading.

As such, in view of significant limitations, I am afraid I cannot recommend publication at this time. In addition, there is a preprint on bioRxiv with seemingly more thorough and compelling analyses than the current manuscript (Fraser et al., 2019: https://www.biorxiv.org/content/10.1101/673665v1).

Major concerns:

1) The data collection process introduced serious selection bias. First, the manuscript uses a non-standard approach to recognizing research articles on PubMed, instead of using the "publication type" field. By using the "date received" and "date published" fields, this procedure immediately excludes dozens of journals that do not publish those dates, such as PNAS, which also happens to rank 6th in terms of total number of preprints published. Second, it is unclear why the manuscript only considers journals that have published at least 50 preprints. This decision introduces selection bias because journals publishing more articles will proportionally have more preprints and journals in certain fields publish more articles/preprints than others. Indeed, a quick look through Table 1 confirms that this analysis only includes articles from very large journals (e.g. PLoS journals) or from fields in which preprints are very popular (e.g. neuroscience and genetics). Third, it is unclear what the manuscript considers 'life sciences', what journals were initially eligible and what journals were excluded because of this definition – for example, what percentage of the articles published by a journal have to be "non-life-science" for it to be excluded?

2) Multiple problems with descriptive and regression analyses. First, it is impossible to appreciate and interpret the findings of the regression analyses without descriptive statistics. The manuscript has to provide a table with descriptive statistics about all covariates included in each regression (in terms of median and interquartile range) as well as a univariable test for each (e.g. p-values from a non-parametric test). Second, such count data are notoriously skewed. As such, even though the log-transformation attempts to capture this skewness, the confidence intervals and p-values may still be wrong. I recommend that the authors instead use bootstrapped confidence intervals and p-values, which can be calculated using the confint(model, method = "boot", boot.type = "basic", nsim = 2000) function of the lme4 package. I also recommend that the manuscript (a) uses a Negative Binomial regression instead of the log-transformation of the response and (b) reports on the diagnostic procedures used to confirm appropriate fit (e.g. by investigating Pearson residuals). The manuscript did well in presenting Figure S8 to illustrate effects without adjusting for the principal components (PCs), the number of which in relation to the number of preprints was quite large (the 1 covariate per 10 outcomes rule of thumb was violated in about ~ 70% of the journals (18/26)), to confirm the apparent absence of overfitting.

3) Interpretation of effect size is in relative rather than absolute terms. When presented, the size of association is interpreted in relative terms (e.g. 1.53 times), instead of absolute terms (e.g. a difference of 2 in median attention score, from 20 to 22). Relative terms are less meaningful and tend to unduly exaggerate the effects estimated. I recommend (a) that the authors present all measures of association (unlike Table 2, which only presents t-statistics) and (b) that all relative terms are either replaced or accompanied by absolute terms; here is an excellent guide on how to do this: https://www.healthnewsreview.org/toolkit/tips-for-understanding-studies/absolute-vs-relative-risk/. I also recommend that any talk of "statistically significant" or "not significant" is replaced by the magnitude of association, as this is what truly matters, and statistical significance language is often confusing and misleading to readers.

4) Inadequate adjustment for scientific field. Even though the authors correctly identify that adjusting for scientific field is hard, the PCA does not convincingly address this concern. First, the approach of using a fixed number of PCs for each journal, rather than a fixed percent of variance explained, means that in half of the journals (13/26) the PCs only explain 50% of the variance due to scientific field or less. Second, the approach of refitting the PCA within each journal, means that even though there was an attempt to account for within-journal variability in scientific field, the between-journal variability is not being accounted for. Third, because of these points, the meta-regression results in a messy estimate of effect from the combination of heterogeneous values (as seen in Figure 1) emanating from regressions adjusting for different study fields to different extends (this heterogeneity was never quantified). The manuscript could address these issues by (a) using a sensitivity analysis to explore the impact of adjusting for different numbers of PCs, (b) using previously published methods to account for scientific field (e.g. Piwowar et al., 2018) or (c) matching articles for subject field using a chosen measure of distance (e.g. Mahalanobis distance) and only using pairs within a pre-specified maximum distance from each other.

5) Lacking in principles of good scientific practice. Even though the authors should be commended for making their data and code available in a neatly-put ZIP file on figshare as well as making their article available as a preprint on bioRxiv, the manuscript would significantly benefit from the following additional practices: (a) make the protocol of this study openly available on either figshare or OSF Registries (https://osf.io/registries), (b) abide by and cite the STROBE guidelines for reporting observational studies (http://www.equator-network.org/reporting-guidelines/strobe/) and (c) include at least a statement on their sources of funding.

6) Poor reporting. This manuscript could derive significant benefit from (a) further and more comprehensive explanation of its methods (e.g. why the choice of 50 or 200, why use regressions followed by meta-regression instead of a random effects model to start with, why use log-transformation instead of Negative Binomial, why use the quoted type of meta-regression, why use the current covariates and not more/less, etc.), (b) avoiding language that may not be familiar to many readers (e.g. fold-change, population stratification, citations + 1, etc.) and (c) adding explanations to figures in the supplement (e.g. what do Figure 1—figure supplement 6 and Figure 1—figure supplement 7 tell us about the PCs, etc.). I actually had to read the Results in combination with the Materials and methods a couple of times to understand that a different regression was fitted for each journal.

[Editors' note: below is the decision letter sent after the second round of peer review.]

Thank you for submitting the revised version of "Releasing a preprint is associated with more attention and citations for the peer-reviewed article" to *eLife*. The revised version has been reviewed by two of the three reviewers who reviewed the previous version. The following individuals involved in review of your submission have agreed to reveal their identity: Olavo Amaral (Reviewer #1).

The reviewers have discussed the reviews with one another and myself, and we would like you to submit a revised version that addresses the points raised by the reviewers (see below). In particular, it is important that the datasets are better described so that other researchers can use them (see points 1 and 3 from Reviewer #1). Reviewer #1 also asks for some further analyses: these are optional – please see below for more details.

Reviewer #1:

The manuscript has been extensively revised and some of my main issues with it have been solved. In particular, methodology (especially article inclusion criteria) is much better described, and data visualization has been improved in many of the figures.

1) However, I still have some issues with data presentation, in particular concerning the supplementary files in which much of the data requested by reviewers has been included. These tables are essentially datasets in. csv format with no legend or clear annotation for the meaning of each column, which is not always obvious from the variable name. Although inclusion of this material is laudable in the sense of data sharing, if the authors mean to use them as a meaningful way to present the results cited in the text, I feel that it is unfair to leave the burden of understanding and analyzing the data on the reader. If they are meant as tables in a scientific paper, it is the author's job to synthesize the data and make them clear to the reader through formatting and annotation, including making variable names self-explanatory and providing legends.

Other general concerns involving analysis are described below:

2) Why is "affiliation in the US" the only geographical factor analyzed? This is rather US-centric, and does not really capture the vast differences between countries in the "non-US" category. Can't the authors make a more meaningful division – for example, based on region/continent, or of economical/scientific development of the country of affiliation?

Note from editor: Please either perform this extra analysis or explain why "affiliation in the US" is the only geographical factor analyzed.

3) I still can get no intuitive meaning of what each of the principal components stand for, and cannot evaluate whether they indeed capture what they mean to (e.g. scientific subfield). The authors do provide a supplementary file with the PC loading, but as the other supplementary files, it is pretty much the raw data, and don't think it's fair for the reader to have to mine it on its own to look for meaning. Can't the authors provide a list of the top MeSH terms loading onto each principal component (as a word cloud, for example), so as to make the meaning of each of them somewhat intuitive?

4) Moreover, if I understood correctly, the principal components are calculated separately for each journal – thus, their meaning varies from one journal to the next. Although that might increase their capability of capturing subfield-specific distinctions, this probably increases the potential that they capture noise rather than signal, due both to sample size decrease and to a decrease in meaningful variability within individual journals. Wouldn't it be more interesting to define subfields based on principal components for the whole sample? Note that this would have the added bonus of allowing inclusion of subfield in the metaregression analysis, and would probably facilitate visualization of the main factors loading onto each component, which would no longer be journal-dependent.

Note from editor: Please either perform this extra analysis or address the comments above about the consequences of the principal components being journal-specific.

5) I very much miss a table of descriptives and individual univariate associations for each variable included in the model before the data on multivariate regression are presented (as mentioned by reviewer #3 in their comments on the first version). Once again, I don't think that throwing in the raw data as a supplementary file substitutes for that.

6) If the authors used time since publication as one of the variables in the model, why didn't they directly test the interaction between this and having a preprint to see whether the relationship changes over time, rather than not doing it and discussing it in the limitations? I understand that there might be confounders, as the authors appropriately discuss in the response to reviewers. However, I feel that discussing the results for the interaction, taking into account the possible confounders and limitations, is still likely to be more interesting than discussing the limitations without a result.

Note from editor: Performing this extra analysis is optional.

Reviewer #2:

The revised manuscript includes a thorough response to the initial comments from reviewers. I believe the analysis has been much improved, and the manuscript now more clearly addresses the concerns that could not be practically addressed. There are only have a few points that could benefit from elaboration within the text.

Introduction, first paragraph: A concern in the previous review was that the statement regarding the proportion of papers that were preprinted was not supported by the provided citation. Though the authors state they have clarified the statement regarding "the number of preprints released [...] is only a fraction of the number of peer reviewed articles published," it remains true that the cited paper says nothing about the overall number of published papers. If the authors want to include a statement about the proportion of published papers, I would point them toward a dataset such as this one, which may provide an acceptable estimate: Penfold NC, Polka J. (2019). Preprints in biology as a fraction of the biomedical literature (Version 1.0) [Data set]. Zenodo. http://doi.org/10.5281/zenodo.3256298

Materials and methods, subsection “Quantifying the associations”: It's still not clear why a spline was used to find the publication date instead of, say, the number of days since 1 Jan 2015. I'm not disputing that it's an appropriate way to encode the dates, but elaboration, as mentioned by a previous reviewer, would be helpful for people like me who have not explicitly encountered this technique before.

Discussion: A previous review comment was that authors with large social media followings may be confounding the analysis by giving themselves a publicity advantage that wasn't included in the analysis. The authors state in their response, "given the weak correlation between Attention Score and citations, it seems unlikely this could explain the effect of preprint status on citations." This is a key point and an interesting rebuttal to the initial suggestion, but I don't believe it's made clear in the paper itself, which says only that online popularity "would likely not explain the association with citations." The manuscript would benefit from clarification here to point out that there is only a loose connection between Attention Score and citations.

Table 1 and Table 2: I believe the advice for multiple-test correction in my earlier review was misguided, I apologize. Though Table 1 now includes adjusted p-values, I'm confused by the approach taken. For 5 of the given p-values, the adjusted value is 2p, while the other 5 have identical adjusted values. Can the authors please check if these values are typos: if they are not, I would suggest they consult a statistician about this analysis, and also about the analysis in Table 2.

---

## [Author Response]

[Editors’ note: this is the author response to the decision letter sent after the first round of peer review.]

Reviewer #1:[…]My primary concern is the large number of confounders that are unaccounted for in the study. Though the authors make a convincing case that the publications with preprints do have an advantage in citations and attention score, there is little attention given in the analysis to the numerous factors that have already been linked to increased attention or citations, factors that may be the actual driver of this effect, rendering preprint status irrelevant.For example, perhaps (published) preprints are longer than articles without preprints [1,2] or have more authors [2]. The results could be affected if senior authors with other highly cited papers are more likely to post a preprint [3,4], or if bioRxiv has an overrepresentation of researchers from "elite" institutions [5] or large research groups [6]. Or maybe authors enthusiastic about preprints also just happen to be more active on Twitter.In short, I am unsure of the utility of measuring complex dynamics like citation count and attention score without incorporating any data about authors.Most importantly, I do not believe this work provides adequate support for the statement in its abstract that "this observational study can help researchers and publishers make informed decisions about how to incorporate preprints into their work." This assertion is directly contradicted by the Discussion section, which accurately assesses the shortcomings of the study. I sympathize with the difficulties in obtaining data on authors, and I do not believe those shortcomings could be remediated in the two-month revision timeline given in the review instructions. I leave it to the discretion of the editors whether this represents an incremental improvement significant enough for publication.

Thank you for the constructive feedback. We have now added several variables to the model to reduce the possibility of confounding. Thus, for each peer-reviewed article, we also include:

- number of authors;

- number of references;

- whether any author had an affiliation in the U.S.;

- amount of time since the last author’s first last-author publication.

All these variables are positively associated with Attention Score and citations, except surprisingly, the last one. Importantly, even after adding these variables to the model, the effect size of releasing a preprint is just as strong. We believe this revision addresses the primary shortcoming in our original submission and improves the credibility of our results. We have revised the manuscript accordingly, including adding the appropriate references to prior work.

It could certainly be true that authors enthusiastic about preprints also just happen to be more active on Twitter, which could partly explain the effect of preprint status on Attention Score. However, given the weak correlation between Attention Score and citations, it seems unlikely this could explain the effect of preprint status on citations. We have added this point to the Discussion.

Additional commetns:Results, first paragraph: I believe there are several ways for this finding to be strengthened, some of which pose significant problems as presented currently:- Most straightforwardly, I don't believe the statement accurately reflects what is actually being measured: "Across journals, each article's Attention Score and citations were weakly correlated" does not convey that the metrics within each journal, and then again within individual years, are, on average, weakly correlated. Clarification would be helpful.

We have simplified and clarified this calculation to be the Spearman correlation between Attention Score and citations within each journal, ignoring time as a variable. The two metrics are still only weakly correlated, and we do not believe a p-value is necessary to make this point. We were originally trying to deal with the fact that both metrics could vary over time, but this turned out to be an unnecessary complication.

*- This result would also benefit from elaboration, here or in the Materials and methods section, regarding why the data were segmented this way to begin with. While other results indicate that journal and year both influence the* difference *between preprinted and non-preprinted publications, is there any evidence that journal and/or year influence the relationship between citations and attention score overall? Comparing all articles together, it appears citations and attention score are correlated with Spearman's rho=0.32. Is there any benefit to instead separating the articles into "journal-years," calculating separate rho values, and finding the median is instead 0.29? If it is to measure the consistency of the relationship across years and journals, this should be stated more explicitly.*

See response above.

- The reported median Spearman correlation and the contents of Figure S2 exclude any consideration of the significance of the correlations, which, while it may not substantially affect the median, does obscure how frequently a journal-year does not have a significant correlation between citations and attention score. A quick estimation using the p-values from the "cor.test" function, corrected for performing 103 correlation tests, suggests there are 16 journal-years that have no significant correlation-changing those rho values to 0 makes Figure S2 look very different, and the largest bin becomes the one indicating no correlation at all. That said, I don't have the knowledge to say if the assumptions used to generate p-values in that package are appropriate here, since the end result is a median of correlations. At the least, if the authors keep the median Spearman correlation, this finding would benefit from performing a permutation test to help us understand whether this median is any better than would be expected at random: Scrambling the citation numbers (possibly within each journal?) and calculating a new median, say, 1,000 times, may help reveal how unexpected a median rho of 0.29 actually is.

See response above.

*Results, second paragraph: I'm curious about the reasoning given for using a log transformation here: "since both metrics were greater than or equal to zero and spanned orders of magnitude." While those are reasonable tests to see whether it's* possible *to log-transform the data, it seems like the actual reason was because it enables using the regression to discuss log-fold change rather than differences in absolute measurements. This statement would benefit from clarification, or a citation supporting why the stated criteria is appropriate.*

We have expanded and clarified our reasoning for the regression modeling. Indeed, one reason we used log-linear regression is that it allowed us to compare the journal-wise log fold-changes, which are on a relative scale.

More importantly, log-linear regression gave the best fit to the data. We have now included direct comparisons of log-linear regression, Gamma regression with a log link, and negative binomial regression (the last one only for citations, since Attention Scores are not necessarily integers). For both metrics and for all journals, and comparing all models on the original scale of the respective metric, log-linear regression had the smallest mean absolute error and mean absolute percentage error.

*Table 2: The authors state that they present uncorrected p-values here because "for each metric, the three variables were tested in one model." This is true, and a nice benefit of meta-regression. However, the paper describes the results of two* different *meta-regression models, which test two different hypotheses (one regarding attention score, and another regarding citations). Though a p-value threshold is not specified for these tests (Results, final paragraph, states only "significantly associated"), convention would suggest the cutoff was 0.05. A Bonferroni correction (for two tests) would push all p values in Table 2 above this threshold-the authors should correct these measurements and update the findings described in the final paragraph of the Results accordingly. Alternatively, it would be acceptable to leave these values as-is and provide further support for not correcting the p-values generated by multiple meta-regression analyses.*

Point taken. We have purged the manuscript of all language related to statistical significance, and added Bonferroni-Holm correction where necessary.

References:(Not intended as a request for the authors to include citations to these papers, just offered here to support my feedback above.)1) Falagas et al. 2013. The Impact of Article Length on the Number of Future Citations: A Bibliometric Analysis of General Medicine Journals. PLOS ONE. doi: 10.1371/journal.pone.0049476.2) Fox et al. 2016. Citations increase with manuscript length, author number, and references cited in ecology journals. Ecology and Evolution. doi: 10.1002/ece3.2505.3) Fu and Aliferis. 2008. Models for Predicting and Explaining Citation Count of Biomedical Articles. AMIA Annual Symposium Proceedings. PMID: 8999029.4) Perc. 2014. The Matthew effect in empirical data. Journal of the Royal Society Interface. doi: 10.1098/rsif.2014.0378.5) Medoff. 2007. Evidence of a Harvard and Chicago Matthew Effect. Journal of Economic Methodology. doi: 10.1080/13501780601049079.6) van Raan. 2006. Performance‐related differences of bibliometric statistical properties of research groups: Cumulative advantages and hierarchically layered networks. Journal of the American Society for Information Science and Technology. doi: 10.1002/asi.20389.Reviewer #2:[…] My major concerns and suggestions are the following:1) Although the statistics used, based on multivariate linear regression, are generally solid, the visualization of results is frequently less intuitive than it could be, and seems to be aimed at the data scientist more than at the average reader. In summary:- In Figure 1A, although the forest plot is a great way to show how results vary by journal, it does not allow the reader to appreciate the range of variability of citations among articles with and without preprints (as the confidence intervals shown depend both on variability and on sample size). As the article is ultimately the unit of analysis here, I miss a standard scatter plot showing unit-level citations for articles with and without preprints – either for the whole set of articles or divided by journal (although this might be too large and better left to a supplementary figure).

Thank you for the feedback. We have revised the figures for clarity. Among other changes, we have added a plot of expected Attention Score and citations for articles with and without a preprint in each journal. In the main text we show this plot with confidence intervals, which do get smaller as sample size increases. In the supplement we show the same plot with prediction intervals, which account for article-to-article variability and do not get smaller as sample size increases. Confidence intervals correspond to the estimate of a population mean, whereas prediction intervals correspond to the estimate of an individual observation.

We have explored various versions of a scatterplot, but the large number of articles, even within one journal, make it uninterpretable. The other advantage of the prediction interval is that it accounts for the other variables that we have now incorporated into the model, which a scatterplot would not.

- Again in terms of visualizing variability, are the authors sure about using log scales in the figures? Although logarithmic transformation is useful for statistics, it is once again not the most intuitive way for the reader to get a sense of the full range of variability.

We believe the revised figures have largely addressed this issue. Because the Attention Scores and citations span orders of magnitude across journals, using a linear scale would highly compress the data points for all but a few journals. The log scale makes it possible to visualize the results for each journal relatively fairly.

- Some of the supplementary figures (particularly Figures S3 and S5) are quite hard to read and understand due to the sheer volume of data presented and might not be very useful to the reader.

We have revised the supplementary figures. In some cases, we have moved the information to supplementary files.

2) Did the author evaluate the interaction between preprints and publication year in the model? This seems like an important question to better assess the likelihood of causality between preprint availability and citations – for more recent articles, I would expect the preprint-related advantage to be greater, as the preprint-only period will account for a larger part of the paper's lifetime. Over time, this advantage is likely to taper off. On the other hand, if an interaction between publication year and preprint availability is not observed, this might suggest that the citation advantage is not necessarily due to the preprint's visibility, but rather to other features or confounders (e.g. articles with preprints being of higher interest, or stemming from groups with more visibility).

Although we agree this is a fascinating suggestion, we are reluctant to perform such an analysis for a couple reasons. First, our data have a relatively small number of preprints and statistical interactions can be difficult to estimate reliably, especially because the rapid growth of preprints in the life sciences means that the majority of preprints in our dataset are linked to newer rather than older peer-reviewed articles. Also, because we encode publication date as a spline with three degrees of freedom, we would be adding three terms to the model, not just one.

Second, we believe the result would be difficult to interpret, because there are multiple factors that are difficult to disentangle. As you suggest, it’s possible that the advantage could taper off on a longer time scale (although the precise time scale is unclear). However, we also know that the advantage at the time of publication is zero (since all articles start at 0 Attention Score and citations). A single linear interaction would not capture non-monotonicity. Another issue is that because we only have Attention Score and citations at one moment in time (the time at which we queried the APIs), the oldest peer-reviewed articles with preprints in our dataset are also the ones published when preprints in the life sciences were just starting to take off, and thus may systematically differ from those with preprints published more recently.

The CrossRef API does not yet provide historical trends of number of citations, and the Altmetric API does not provide sufficient temporal resolution for historical Attention Score.

3) Similarly, it would be interesting to investigate whether metrics of the preprint such as altmetric scores, number of downloads, or even citations – all of which are available in bioRxiv or CrossRef) are related to the citation advantage for their published versions. Although showing that the advantage is larger for preprints with more visibility does not prove that posting a preprint leads to more citations, finding that the advantage does not correlate with attention received to the preprint would argue against it. All of these could be performed as secondary analyses similar to the one performed for the number of days the preprint preceded the peer-reviewed article.

Here we encounter a similar difficulty. To do such an analysis fairly and avoid analyzing positive feedback loops (e.g., in which the preprint gets attention and citations due to the peer-reviewed article), we would want Attention Scores and number of citations prior to publication of the peer-reviewed article. Because we lack sufficient historical data, we have left this analysis for future work.

4) The criteria for journal selection should be explained more clearly for the reader to assess the sample's representativeness. Moreover, the principal components used in subfield analysis could be better described (explicitly showing the terms with high loading in each of them) for one to get a sense of how meaningful they are to define subfields. Both of these themes are expanded in the minor comments.

We have revised and clarified the inclusion criteria for peer-reviewed articles and journals. We have also moved the PC loadings to a supplementary file, where they are easier to read.

Reviewer #3:[…]The authors should be commended for attempting to study and promote the advantages of preprints, validate and extend previous work, for openly providing their data and code and for including a good limitations section. However, the chosen method of data collection is highly prone to selection bias, many of the descriptive and analytic choices are poorly justified and the magnitude of association, wherever presented, is quantified in relative rather than absolute terms, which in the setting of highly skewed data is very misleading.As such, in view of significant limitations, I am afraid I cannot recommend publication at this time. In addition, there is a preprint on bioRxiv with seemingly more thorough and compelling analyses than the current manuscript (Fraser et al., 2019: https://www.biorxiv.org/content/10.1101/673665v1).

Thank you for your feedback. We have thoroughly revised the analysis and manuscript in response to your concerns. We appreciate the study by Fraser et al. We believe our work has several strengths, and that the two studies complement each other well.

Major concerns:1) The data collection process introduced serious selection bias. First, the manuscript uses a non-standard approach to recognizing research articles on PubMed, instead of using the "publication type" field. By using the "date received" and "date published" fields, this procedure immediately excludes dozens of journals that do not publish those dates, such as PNAS, which also happens to rank 6th in terms of total number of preprints published. Second, it is unclear why the manuscript only considers journals that have published at least 50 preprints. This decision introduces selection bias because journals publishing more articles will proportionally have more preprints and journals in certain fields publish more articles/preprints than others. Indeed, a quick look through Table 1 confirms that this analysis only includes articles from very large journals (e.g. PLoS journals) or from fields in which preprints are very popular (e.g. neuroscience and genetics). Third, it is unclear what the manuscript considers 'life sciences', what journals were initially eligible and what journals were excluded because of this definition – for example, what percentage of the articles published by a journal have to be "non-life-science" for it to be excluded?

Thank you for this suggestion. We were using the “publication type” field before, but are now using it more extensively. We have revised our inclusion criteria for articles and journals, and clearly described them in the Materials and methods. Revising the inclusion criteria increased the size of our dataset to ~74,000 peer-reviewed articles.

We selected the cutoff for number of articles released as preprints based on the number of variables in the regression models (Austin and Steyerberg, 2015). Because our analysis is stratified on journal, this ensures stable estimates of the model coefficients. We acknowledge that our results only apply to journals that have published a minimum number of articles previously released as preprints.

We excluded multidisciplinary journals because many articles published in these journals are unlikely to be released on bioRxiv, which could confound the analysis. We originally identified these journals manually, but we have now verified them using the categories in the Journal Citation Reports published by Clarivate Analytics.

2) Multiple problems with descriptive and regression analyses. First, it is impossible to appreciate and interpret the findings of the regression analyses without descriptive statistics. The manuscript has to provide a table with descriptive statistics about all covariates included in each regression (in terms of median and interquartile range) as well as a univariable test for each (e.g. p-values from a non-parametric test). Second, such count data are notoriously skewed. As such, even though the log-transformation attempts to capture this skewness, the confidence intervals and p-values may still be wrong. I recommend that the authors instead use bootstrapped confidence intervals and p-values, which can be calculated using the confint(model, method = "boot", boot.type = "basic", nsim = 2000) function of the lme4 package. I also recommend that the manuscript (a) uses a Negative Binomial regression instead of the log-transformation of the response and (b) reports on the diagnostic procedures used to confirm appropriate fit (e.g. by investigating Pearson residuals). The manuscript did well in presenting Figure S8 to illustrate effects without adjusting for the principal components (PCs), the number of which in relation to the number of preprints was quite large (the 1 covariate per 10 outcomes rule of thumb was violated in about ~ 70% of the journals (18/26)), to confirm the apparent absence of overfitting.

We have added a supplementary file of descriptive statistics for each variable in the model. Because each variable was associated with either Attention Score or citations at a nominal level by random-effects meta-analysis, we have not included the results of univariate tests.

For both metrics and for all journals, we have run log-linear regression and Gamma regression (latter with a log link). For citations, we have also run negative binomial regression. Attention Scores are not limited to integer values, and so are not appropriate for negative binomial regression. Comparing the fits from log-linear, Gamma, and negative binomial regression on the original scale of the respective metric, log-linear regression had the smallest mean absolute error and mean absolute percentage error for each metric and each journal. We also used the function glm.diag.plots to examine the distributions of residuals from each method. We believe these results establish the validity of our regression approach. We have revised the manuscript accordingly.

We initially considered fitting a mixed-effects model based on all the data, but decided to instead stratify our analysis by journal. Our primary concern is that a mixed-effects model would make unreasonably strong assumptions about the distribution of effects for a given variable (e.g., publication date or a MeSH term PC). Since we have not fit a mixed-effects model, we have not used lme4::confint.merMod.

3) Interpretation of effect size is in relative rather than absolute terms. When presented, the size of association is interpreted in relative terms (e.g. 1.53 times), instead of absolute terms (e.g. a difference of 2 in median attention score, from 20 to 22). Relative terms are less meaningful and tend to unduly exaggerate the effects estimated. I recommend (a) that the authors present all measures of association (unlike Table 2, which only presents t-statistics) and (b) that all relative terms are either replaced or accompanied by absolute terms; here is an excellent guide on how to do this: https://www.healthnewsreview.org/toolkit/tips-for-understanding-studies/absolute-vs-relative-risk/. I also recommend that any talk of "statistically significant" or "not significant" is replaced by the magnitude of association, as this is what truly matters, and statistical significance language is often confusing and misleading to readers.

We have purged the manuscript of all mention of statistical significance. We have revised the figures to show absolute effects along with relative effects for each journal. We believe the relative effects are useful for comparing across journals, and they are the natural output of log-linear regression, which gave the best fit to the data. We have included all measures of association in the revised tables.

4) Inadequate adjustment for scientific field. Even though the authors correctly identify that adjusting for scientific field is hard, the PCA does not convincingly address this concern. First, the approach of using a fixed number of PCs for each journal, rather than a fixed percent of variance explained, means that in half of the journals (13/26) the PCs only explain 50% of the variance due to scientific field or less. Second, the approach of refitting the PCA within each journal, means that even though there was an attempt to account for within-journal variability in scientific field, the between-journal variability is not being accounted for. Third, because of these points, the meta-regression results in a messy estimate of effect from the combination of heterogeneous values (as seen in Figure 1) emanating from regressions adjusting for different study fields to different extends (this heterogeneity was never quantified). The manuscript could address these issues by (a) using a sensitivity analysis to explore the impact of adjusting for different numbers of PCs, (b) using previously published methods to account for scientific field (e.g. Piwowar et al., 2018) or (c) matching articles for subject field using a chosen measure of distance (e.g. Mahalanobis distance) and only using pairs within a pre-specified maximum distance from each other.

We have tried to balance the number of PCs against the number of preprints per journal. For linear regression, previous work suggests that having as few as two observations per independent variable is sufficient to reliably estimate the coefficients (Austin and Steyerberg, 2015). This differs from the rule of thumb for logistic regression, which is ~10 observations per independent variable.

We have increased the number of MeSH PCs from 10 to 15 and obtained similar results. We have also included tables of all regression statistics for all variables. For a given journal, typically only a couple PCs are associated with either Attention Score or citations.

We intentionally calculated the PCs separately for each journal, because the MeSH terms associated with more citations in one journal could be irrelevant or associated with fewer citations in another journal. This is one reason we chose not to fit one mixed-effects model from all the data, as it would make strong assumptions about the distribution of effect sizes for a given PC. We believe this is a worthwhile tradeoff for performing meta-regression on the coefficients for preprint status.

Piwowar et al. only applied their article-by-article classification approach to articles published in multidisciplinary journals, which we have specifically excluded. In addition, their approach requires detailed information on the references for a given article, which does not exist for every article in our dataset. Matching would require another arbitrary parameter (e.g., maximum distance), would be complicated to incorporate alongside the other variables we have added to the model, and would likely also reduce our sample size, even if we attempted to match multiple articles without a preprint to one article with a preprint.

5) Lacking in principles of good scientific practice. Even though the authors should be commended for making their data and code available in a neatly-put ZIP file on figshare as well as making their article available as a preprint on bioRxiv, the manuscript would significantly benefit from the following additional practices: (a) make the protocol of this study openly available on either figshare or OSF Registries (https://osf.io/registries), (b) abide by and cite the STROBE guidelines for reporting observational studies (http://www.equator-network.org/reporting-guidelines/strobe/) and (c) include at least a statement on their sources of funding.

We have thoroughly revised our descriptions of the selection criteria for articles and journals and the source of each variable in the model. We have also added a funding statement, which we had mistakenly omitted. As far as we understand them, the STROBE guidelines are meant for studies on humans.

6) Poor reporting. This manuscript could derive significant benefit from (a) further and more comprehensive explanation of its methods (e.g. why the choice of 50 or 200, why use regressions followed by meta-regression instead of a random effects model to start with, why use log-transformation instead of Negative Binomial, why use the quoted type of meta-regression, why use the current covariates and not more/less, etc.), (b) avoiding language that may not be familiar to many readers (e.g. fold-change, population stratification, citations + 1, etc.) and (c) adding explanations to figures in the supplement (e.g. what do Figure 1—figure supplement 6 and Figure 1—figure supplement 7 tell us about the PCs, etc.). I actually had to read the results in combination with the methods a couple of times to understand that a different regression was fitted for each journal.

We have heavily revised the text to more clearly explain various aspects of the analysis. Please see our responses above for specific justifications.

[Editors' note: this is the author response to the decision letter sent after the second round of peer review.]

Reviewer #1:The manuscript has been extensively reviewed and some of my main issues with it have been solved. In particular, methodology (especially article inclusion criteria) is much better described, and data visualization has been improved in many of the figures.1) However, I still have some issues with data presentation, in particular concerning the supplementary files in which much of the data requested by reviewers has been included. These tables are essentially datasets in. csv with no legend or clear annotation for the meaning of each column, which is not always obvious from the variable name. Although inclusion of this material is laudable in the sense of data sharing, if the authors mean to use them as a meaningful way to present the results cited in the text, I feel that it is unfair to leave the burden of understanding and analyzing the data on the reader. If they are meant as tables in a scientific paper, it is the author's job to synthesize the data and make them clear to the reader through formatting and annotation, including making variable names self-explanatory and providing legends.

Thank you for the suggestion. We had originally included the supplementary files exactly as generated by the analysis code. We have now renamed the columns to be more interpretable, and added a description of each table in the manuscript file. We much prefer to keep them in plain text format (csv instead of xlsx) to prevent Excel from re-formatting the dates and numbers.

Other general concerns involving analysis are described below:2) Why is "affiliation in the US" the only geographical factor analyzed? This is rather US-centric, and does not really capture the vast differences between countries in the "non-US" category. Can't the authors make a more meaningful division – for example, based on region/continent, or of economical/scientific development of the country of affiliation?Note from editor: Please either perform this extra analysis or explain why "affiliation in the US" is the only geographical factor analyzed.

Although the binary variable of having a U.S. affiliation is admittedly crude, the U.S. was by far the most common country of affiliation in our dataset. In addition, according to Adbill and Blekhman 2019, “the majority of the top 100 universities (by author count) [of preprints on bioRxiv] are based in the United States”. We were also trying to avoid adding too many variables to the model.

We have now added a binary variable of having an affiliation at an institution in the Nature Index for Life Sciences. These 100 institutions have the highest number of articles published in the “upper echelon” of life science journals, and thus are a proxy for the world’s “elite” institutions (see revised Materials and methods section for details). Although affiliation at a Nature Index institution is associated with higher Attention Score and more citations, adding that variable to the model did not markedly change the coefficients for having a preprint. We have revised the manuscript accordingly.

3) I still can get no intuitive meaning of what each of the principal components stand for, and cannot evaluate whether they indeed capture what they mean to (e.g. scientific subfield). The authors do provide a supplementary file with the PC loading, but as the other supplementary files, it is pretty much the raw data, and don't think it's fair for the reader to have to mine it on its own to look for meaning. Can't the authors provide a list of the top MeSH terms loading onto each principal component (as a word cloud, for example), so as to make the meaning of each of them somewhat intuitive?

We have replaced the previous supplementary file of PC loadings with a file of the MeSH terms with the largest positive and negative loadings for each PC (Supplementary File 7). Since we have 39 journals and 15 PCs, we believe this is more practical than a word cloud.

4) Moreover, if I understood correctly, the principal components are calculated separately for each journal – thus, their meaning varies from one journal to the next. Although that might increase their capability of capturing subfield-specific distinctions, this probably increases the potential that they capture noise rather than signal, due both to sample size decrease and to a decrease in meaningful variability within individual journals. Wouldn't it be more interesting to define subfields based on principal components for the whole sample? Note that this would have the added bonus of allowing inclusion of subfield in the metaregression analysis, and would probably facilitate visualization of the main factors loading onto each component, which would no longer be journal-dependent.Note from editor: Please either perform this extra analysis or address the comments above about the consequences of the principal components being journal-specific

You understood correctly, we calculate the PCs separately for each journal. We explained some of our reasoning previously to reviewer #3. We do not believe it would be more interesting to define subfields for the entire dataset. Because the journals cover disparate research areas, MeSH terms common in articles for one journal can be non-existent in articles for another. Our approach allows us to calculate PCs that are specifically tuned to the variation in MeSH term assignment between articles in a given journal, rather than having the majority of PCs describe irrelevant, higher-level variation between journals (which would add noise to each regression). Given the large number of articles from each journal (minimum 304 articles, and 25 of 39 journals have ≥ 1000 articles), we do not expect noise in the PCs to be an issue. This is supported by the fact that several PCs for each journal are associated with Attention Score and citations at p ≤ 0.05.

We think it would be statistically ill-advised to incorporate the PCs into both the regression and meta-regression, and we would much rather have them in the regressions. In any case, we do not believe incorporating subfield into the meta-regression would be advantageous, because it would assume, for example, that the effect on Attention Score and citations of publishing a paper about *Saccharomyces cerevisiae* is the same whether one publishes the paper in Molecular Cell or in Neuroimage. Such an assumption does not make sense to us. We have elaborated on our reasoning in the manuscript.

5) I very much miss a table of descriptives and individual univariate associations for each variable included in the model before the data on multivariate regression are presented (as mentioned by reviewer #3 in his comments on the first version). Once again, I don't think that throwing in the raw data as a supplementary file substitutes for that.

We had included a table of descriptive statistics, which we have annotated more thoroughly along with the other supplementary files. We do not believe a supplementary file of the univariate associations of each variable with each metric in each journal would be useful. Our results already show that each variable has a non-zero coefficient in the multivariate models for Attention Score and citations, so the results of univariate testing would not change the subsequent analysis (e.g., we’re not going to remove a variable that we added in response to reviewer comments).

6) If the authors used time since publication as one of the variables in the model, why didn't they directly test the interaction between this and having a preprint to see whether the relationship changes over time, rather than not doing it and discussing it in the limitations? I understand that there might be confounders, as the authors appropriately discuss in the response to reviewers. However, I feel that discussing the results for the interaction, taking into account the possible confounders and limitations, is still likely to be more interesting than discussing the limitations without a result.Note from editor: Performing this extra analysis is optional.

We respectfully disagree. We believe the reasons to not test for an interaction greatly outweigh the reasons to test for one. Given the limitations, we do not want to put ourselves in the position of having to interpret complicated, questionable results that do not alter our study’s conclusions.

Reviewer #2:The revised manuscript includes a thorough response to the initial comments from reviewers. I believe the analysis has been much improved, and the manuscript now more clearly addresses the concerns that could not be practically addressed. There are only have a few points that could benefit from elaboration within the text.Introduction, first paragraph: A concern in the previous review was that the statement regarding the proportion of papers that were preprinted was not supported by the provided citation. Though the authors state they have clarified the statement regarding "the number of preprints released [...] is only a fraction of the number of peer reviewed articles published," it remains true that the cited paper says nothing about the overall number of published papers. If the authors want to include a statement about the proportion of published papers, I would point them toward a dataset such as this one, which may provide an acceptable estimate: Penfold NC, Polka J. (2019). Preprints in biology as a fraction of the biomedical literature (Version 1.0) [Data set]. Zenodo. http://doi.org/10.5281/zenodo.3256298

Thank you for the reference, we have now cited the dataset of Penfold and Polka.

Materials and methods, subsection “Quantifying the associations”: It's still not clear why a spline was used to find the publication date instead of, say, the number of days since 1 Jan 2015. I'm not disputing that it's an appropriate way to encode the dates, but elaboration, as mentioned by a previous reviewer, would be helpful for people like me who have not explicitly encountered this technique before.

We have elaborated on the reason for the spline. Unlike a single linear term such as “number of days since publication”, the spline does not assume, for example, that the average difference in the dependent variable (e.g., log citations) between a 0-year-old article and a 1-year-old article is the same as between a 4-year-old article and a 5-year-old article.

Discussion: A previous review comment was that authors with large social media followings may be confounding the analysis by giving themselves a publicity advantage that wasn't included in the analysis. The authors state in their response, "given the weak correlation between Attention Score and citations, it seems unlikely this could explain the effect of preprint status on citations." This is a key point and an interesting rebuttal to the initial suggestion, but I don't believe it's made clear in the paper itself, which says only that online popularity "would likely not explain the association with citations." The manuscript would benefit from clarification here to point out that there is only a loose connection between Attention Score and citations.

We have clarified that sentence in the Discussion.

Table 1 and Table 2: I believe the advice for multiple-test correction in my earlier review was misguided, I apologize. Though Table 1 now includes adjusted p-values, I'm confused by the approach taken: For 5 of the given p-values, the adjusted value is 2p, while the other 5 have identical adjusted values. Can the authors please check if these values are typos: if they are not, I would suggest they consult a statistician about this analysis, and also about the analysis in Yable 2.

They are not typos, that’s just how the Bonferroni-Holm correction works. One can verify this in R, e.g., “p.adjust(c(1e-3, 1e-4), method = 'holm')”. Bonferroni-Holm is uniformly more powerful than standard Bonferroni (https://en.wikipedia.org/wiki/Holm%E2%80%93Bonferroni_method). We have clarified this in the Materials and methods and the figure legends.